# Non-coding somatic mutations converge on the PAX8 pathway in ovarian cancer

Rosario I. Corona [1,2,13], Ji-Heui Seo[3,4,13], Xianzhi Lin [1,13], Dennis J. Hazelett[2], Jessica Reddy[1], Marcos A. S. Fonseca[1], Forough Abassi[1], Yvonne G. Lin[5], Paulette Y. Mhawech-Fauceglia[6], Sohrab P. Shah [7,8,9], David G. Huntsman[8,9,10], Alexander Gusev [3,11], Beth Y. Karlan[1], Benjamin P. Berman[2], Matthew L. Freedman[3,4,12,14✉], Simon A. Gayther [1,2,14✉] & Kate Lawrenson[1,2,14✉]

The functional consequences of somatic non-coding mutations in ovarian cancer (OC) are unknown. To identify regulatory elements (RE) and genes perturbed by acquired non-coding variants, here we establish epigenomic and transcriptomic landscapes of primary OCs using H3K27ac ChIP-seq and RNA-seq, and then integrate these with whole genome sequencing data from 232 OCs. We identify 25 frequently mutated regulatory elements, including an enhancer at 6p22.1 which associates with differential expression of ZSCAN16 ($P = 6.6 \times 10^{-4}$) and ZSCAN12 ($P = 0.02$). CRISPR/Cas9 knockout of this enhancer induces downregulation of both genes. Globally, there is an enrichment of single nucleotide variants in active binding sites for TEAD4 ($P = 6 \times 10^{-11}$) and its binding partner PAX8 ($P = 2 \times 10^{-10}$), a known lineage-specific transcription factor in OC. In addition, the collection of *cis* REs associated with PAX8 comprise the most frequently mutated set of enhancers in OC ($P = 0.003$). These data indicate that non-coding somatic mutations disrupt the PAX8 transcriptional network during OC development.

[1] Cedars-Sinai Women's Cancer Program at the Samuel Oschin Cancer Center, Los Angeles, CA, USA. [2] Center for Bioinformatics and Functional Genomics, Cedars-Sinai Medical Center, Los Angeles, CA, USA. [3] Department of Medical Oncology, Dana-Farber Cancer Institute, Boston, MA, USA. [4] Center for Functional Cancer Epigenetics, Dana-Farber Cancer Institute, Boston, MA, USA. [5] Department of Obstetrics and Gynecology, Keck School of Medicine, University of Southern California, Los Angeles, CA, USA. [6] Department of Pathology, University of Southern California, Los Angeles, CA, USA. [7] Department of Computer Science, University of British Columbia, Vancouver, BC, Canada. [8] Department of Molecular Oncology, BC Cancer Agency, Vancouver, BC, Canada. [9] Department of Pathology and Laboratory Medicine, University of British Columbia, Vancouver, BC, Canada. [10] Department of Gynecology and Obstetrics, University of British Columbia, Vancouver, BC, Canada. [11] McGraw/Patterson Center for Population Sciences, Dana-Farber Cancer Institute, Boston, MA, USA. [12] The Eli and Edythe L. Broad Institute, Cambridge, MA, USA. [13] These authors contributed equally: Rosario I. Corona, Ji-Heui Seo, Xianzhi Lin. [15] These authors jointly supervised this work: Matthew L. Freedman, Simon A. Gayther, Kate Lawrenson. ✉email: mfreedman@partners.org; simon.gayther@cshs.org; kate.lawrenson@cshs.org

Epithelial ovarian cancer (OC) is a heterogeneous disease, comprising several different histological subtypes that differ in their underlying genetics, epidemiologic risk factors, clinical characteristics, and cellular origins[1–3]. The most common subtype is high-grade serous OC (HGSOC), which accounts for around two-thirds of all invasive OCs, and likely arises from fallopian tube secretory epithelium[4]. Other subtypes of invasive OC are rarer and include clear cell and endometrioid OCs (CCOC and EnOC), which are strongly associated with the benign precursor lesion endometriosis[5], and mucinous OC (MOC), which may derive from appendiceal tissues or primordial germ cells[6].

Molecular analyses of primary OCs have so far focused on characterizing somatic genomic variation in protein-coding genes, which have implicated distinct genetic alterations and biological pathways in the development of each histotype. TP53 mutations are ubiquitous in primary HGSOCs[7]. Loss-of-function mutations in DNA double-strand break repair genes (e.g. BRCA1, BRCA2), which confer high-penetrance genetic susceptibility to OC, are also relatively common[8] and predicate a genomic instability phenotype that results in an accumulation of gross structural genomic changes as tumors develop[9]. CCOCs and EnOCs often harbor somatic pathogenic mutations in ARID1A, a member of the SWI/SNF family of chromatin remodelers[10]. TERT promoter mutations are specific to CCOCs[11], and coding mutations and promoter methylation in DNA mismatch repair genes are relatively common in the EnOC subtype[12]. Alterations in the MAPK pathway are present in ~70% of MOCs, with KRAS hotspot mutations (amino acid 12 or 13) the most frequent genetic change[13].

There is currently a lack of understanding of the role of non-coding mutations in cancer pathogenesis. Whole genome-sequencing (WGS) studies shows that ~96% of all somatic mutations identified in primary tumors lie in non-protein coding DNA regions. A proportion of non-coding somatic mutations likely represent functional drivers of cancer disease development[14–16], mediating their effects by modifying the function of regulatory elements (REs) that modify the expression of target genes that contribute to neoplastic development. The architecture of gene regulation is highly tissue and cell-type specific[17]. Somatic mutations within disease-specific REs are therefore expected to affect gene expression in a disease-specific manner.

The goals of the current study are: (1) To characterize the histotype-specific regulatory landscapes of the different OC histotypes and (2) using WGS data from primary ovarian tumors, to identify frequently mutated REs (FMREs) that may play a role in disease pathogenesis (Fig. 1a).

## Results

**Epigenomic and transcriptomic landscapes of OCs**. We characterized the histotype-specific landscapes of active chromatin in primary ovarian tumors using chromatin immunoprecipitation-sequencing (ChIP-seq) for acetylated lysine 27 of histone H3 protein (H3K27ac). H3K27ac ChIP-seq was performed in 20 primary tumors, 5 each for the different histotypes, HGSOC, CCOC, EnOC, and MOC (Supplementary Table 1). We identified a union set of 295,243 non-overlapping ChIP-seq peaks across all tumors, comprising 11.6% of the genome (Fig. 1b and Supplementary Fig. 1). The number of peaks identified plateaued at 16 tumors, suggesting we have identified the majority of active REs in OCs. The 12,954 peaks (4.4%) shared across all tumor types were enriched at gene promoters (odds ratio = 14, $P < 0.001$, Fisher's exact test, when compared genome-wide) (Fig. 1c). For each tumor type, we also identified a set of histotype-specific H3K27ac peaks: 6,583 in HGSOCs, 5,401 in CCOCs, 2,134 in

MOCs, and 20 in EnOCs. These peaks fall predominantly in enhancers (odds ratio = 40, $P < 0.001$, Fisher's exact test, histotype-specific H3K27ac peaks versus common H3K27ac peaks in OCs) (Fig. 1c).

We performed RNA sequencing (RNA-seq) in 19 of these tumors. Consistent with H3K27ac ChIP-seq data, we identified histotype-specific patterns of gene expression for each tumor type. There were 1,214 differentially expressed genes (DEGs) specific to CCOC, 519 DEGs for HGSOC, 371 DEGs for MOC, and 16 DEGs for EnOC (Supplementary Fig. 2). Patterns of chromatin activity were consistent with patterns of gene expression; genes flanking histotype-specific peaks of active chromatin were consistently expressed at higher levels in tumors of the same histotype. Conversely, lower H3K27ac signal was associated with lower expression of nearby genes (Fig. 1d). As an example, we observed elevated H3K27ac ChIP-seq signal in the promoter of WFDC2 associated with higher expression of the WFDC2 gene in HGSOC and EnOC samples compared to CCOCs and MOCs (Spearman's rho = 0.78, $P = 5.5 \times 10^{-5}$) (Fig. 1e–g). WFDC2 encodes for human epididymis protein 4 (HE4), a biomarker overexpressed in serous and endometrioid OCs[18]. HE4 is used clinically to monitor OC recurrence. Taken together, these data indicate that histotype-specific enhancers regulate gene expression in cis in a histotype-type specific manner.

**Predicting REs and target genes interactions**. We integrated H3K27ac ChIP-seq and RNA-seq data to map REs, including typical enhancers and promoters, to putative target genes. We calculated all correlations between gene expression and RE activity to identify all gene–RE pairs within topologically associating domains (TADs) (Spearman's rho > 0.4, $P < 0.05$, distance <500 kbp) (Fig. 1h). We use the term 'CREAG' (collection of 'cis-Regulatory Elements Associated with a Gene'), to define the collection of predicted enhancers for a gene of interest (a CREAG is conceptually similar to the previously described gene 'plexi'[19]). This defined a catalog of 15,380 RE–gene associations between 6,197 genes and 11,371 REs across all histotypes, with a median of 2.5 enhancers per CREAG (Fig. 1i) and 1.4 genes per enhancer (Fig. 1j). We compared enhancer–gene associations with the GeneHancer database to infer target genes for 285,000 human enhancers[20]. Forty-four percent of our predicted enhancer–gene associations were annotated to the same gene in GeneHancer, seven times more than expected by chance ($P < 0.001$). We then mapped all histotype-specific REs to their putative target gene(s) (Supplementary Data 1) and performed pathway enrichment analysis to identify biological mechanisms associated with each histotype (Fig. 1k and Supplementary Fig. 3). Genes associated with histotype-specific REs function in pathways known to be involved the development of each histotype. RE-associated genes for CCOC (e.g. AKT2, ITGA5, LAMC1, MET, PPP2R3A, and SGK1) are involved in PI3K-Akt signaling ($P = 0.0017$)[3,21,22]; and active REs specific to MOCs are associated with genes (e.g. GCNT3, MUC12, GALNT5 and B3GNT5) involved in O-glycan processing ($P = 0.0001$), likely reflecting upregulated mucin production and O-glycosylation activity in this histotype[23]. Pathways common across histotypes were associated with cell proliferation and mitosis (Supplementary Fig. 3).

**Super-enhancer landscapes in OCs**. Large enhancer domains, termed super-enhancers (SEs) or stretch enhancers[24,25], typically regulate genes critical to cell identity. We asked whether histotype-specific SEs were present, and whether these were associated with genes that may contribute to phenotypic heterogeneity in OCs. We characterized between 653 and 1,945 SEs per

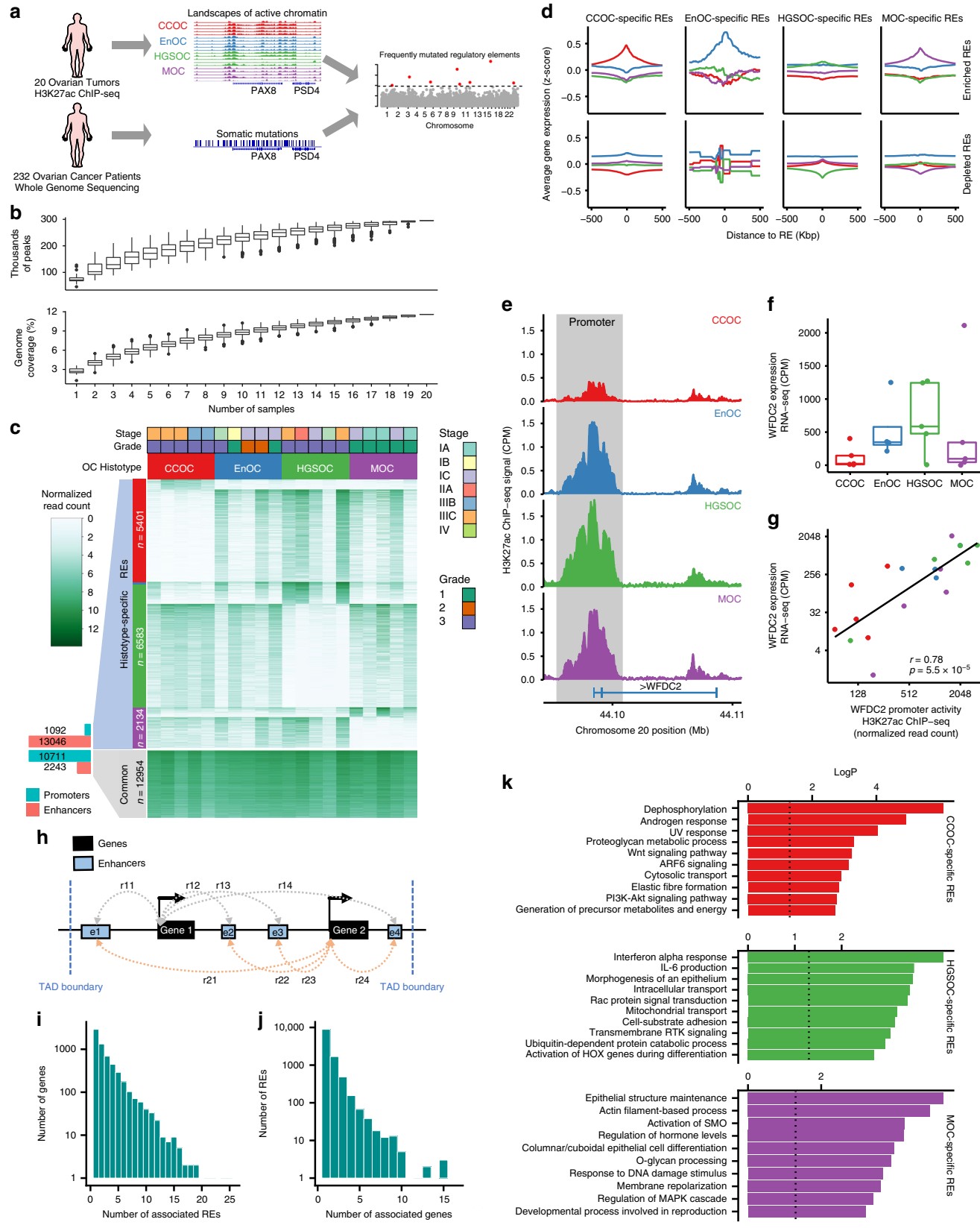

tumor (mean = 1245, sd = 321) (Fig. 2a, Supplementary Table 2). By assigning SEs to the highest expressed overlapping gene we identified a total of 5,338 SE-associated genes, of which 1,123 were common to all OC histotypes. *PAX8*, a transcription factor and known essential gene in OC[26], was associated with SEs in 17

(out of 20) tumors, with the lowest signal in MOCs (Fig. 2b and Supplementary Fig. 4). *MUC16*, which encodes CA125, another clinical biomarker used to diagnose and monitor OC, coincided with an SE in 13/20 tumors; again the lowest enhancer activity was in MOCs, which are known to have the lowest CA125

**Fig. 1 Epigenomic profiling in 20 epithelial OCs reveals histotype-specific REs. a** Study overview— leveraging landscapes of active chromatin in ovarian cancer to identify frequently mutated regulatory elements. **b** Number of peaks and genome coverage as a function of number of samples. **c** Heatmap showing the normalized H3K27ac ChIP-seq signal for the 20 OC samples (columns) at the active REs (rows). **d** Gene expression averaged by OC histotype (CCOC, red; EnOC, blue; HGSOC, green; MOC, purple) around histotype-specific REs. **e–g** *WFDC2* locus; **e** H3K27ac ChIP-seq signal in the promoter region (chr20:44,095,981–44,101,060) and **f** gene expression in different histotypes of OC. **g** Normalized H3K27ac ChIP-seq signal versus *WFDC2* gene expression. **h** Diagram of the enhancer–gene association strategy that computes the Spearman's correlation between enhancer activity (normalized H3K27ac ChIP-seq signal) and gene expression (normalized RNA-seq) ($r_{ij}$) between all enhancers and all genes within the same topologically associating domain (TAD). A putative enhancer–gene association is established if the correlation ($r_{ij}$) is significant ($r_{ij} > 0.4$ and *P*-value < 0.05). **i** Histogram of number of associated genes per RE and **j** number of associated REs per gene. **k** Pathway enrichment analysis of genes associated with histotype-specific REs.

expression of the four major histotypes[27]. SE-associated genes for OC include the highly expressed lncRNA *LINC00963*, which was identified in 17/20 tumors (average expression = 470 CPM, sd = 585 CPM, Supplementary Fig. 3). *LINC00963* has been implicated in prostate cancer progression[28] but has not been previously associated with OC.

A total of 2,094 SE-associated genes were histotype-specific: 634 for CCOC, 617 for HGSOC, 565 for MOC, and 278 for EnOC (Fig. 2c). As expected, histotype-specific SE-associated genes were more highly expressed in the histotype of interest (Supplementary Fig. 4); for example, SE-associated genes specific to CCOC had a median normalized gene expression above zero, but less than zero for all other histotypes. The known function of many of the SE-associated genes we identified support a histotype-specific role in OC (Fig. 2d-f); for example the CCOC-specific gene protein phosphatase 1 regulatory subunit 3B (*PPP1R3B*) regulates glycogen synthesis, consistent with the observations that clear cell tumors contain large deposits of cytoplasmic glycogen. *PPP1R3B* is expressed more highly in CCOC cell lines than in HGSOC and normal FTSEC lines (Fig. 2g). In vitro knockdown of *PPP1R3B* in two CCOC models (JHOC5 and RMG-II cell lines) results in a significant decrease in glycogen content (*P* = 0.05, Fig. 2h and i). SE-associated genes common to all OCs are involved in pathways, such as TNFA signaling via NF-kB and response to growth factor (*P* = $2.7 \times 10^{-25}$ and $1.3 \times 10^{-14}$, respectively (Supplementary Fig. 4).

**Somatically mutated REs in OCs.** We collated whole genome sequencing (WGS) data from 232 primary OCs which included 169 HGSOCs, 28 CCOCs, and 35 EnOCs[3] to identify the somatic non-coding mutations occurring in active enhancers and promoters in OCs. In total, there were 1.7 million single nucleotide variants (SNVs) across all 232 tumors, with an average of 7,163 SNVs per tumor (range 480–40,576) (Supplementary Fig. 5). Of these, 1.6 million (98.8%) SNVs lay in non-coding DNA regions (Supplementary Fig. 5). Approximately 9.3% percent of SNVs are located in active REs. Fourteen percent of these are in active promoters and 86% in active enhancers. We looked for FMREs harboring SNVs at a frequency greater than expected by chance based on the average distribution of mutations throughout all H3K27ac positive regions (see "Methods" section). We identified 25 FMREs across all histotypes at a false discovery rate (FDR) of 0.25 (Fig. 3a–c). Eight FMREs were unique to HGSOC, 17 were unique to endometriosis-associated OCs (CCOC and EnOC) (Supplementary Table 3). FMREs included both promoters and enhancers. To evaluate the functional consequences of FMREs we used our gene–RE maps to quantify global patterns of differential gene expression associated with somatic SNVs in promoters and enhancers. For 89 HGSOCs both WGS and RNA-seq data were available, which enabled us to quantify differential gene expression associated with RE mutation. We found that overall, genes were significantly more likely to be overexpressed in samples with RE mutations compared to wild-type samples, i.e., samples without somatic mutations in the RE of interest. For the 2,893

REs harboring at least 1 somatic SNV, 89 associated genes were significantly overexpressed (FC > 2) and 46 genes were significantly downregulated (FC < 0.5) (*P*-value < 0.05, binomial distribution), indicating that SNVs in enhancers are around twice as likely to activate rather than repress gene expression (Fig. 3d). At chromosome 10p15, we identified a cluster of nine somatic SNVs all located within the *KLF6* promoter, within a common SE in primary OCs (Fig. 3e). Two somatic SNVs coincided with a binding site for PAX8 in the promoter, defined by H3K27ac ChIP-seq in OC cell lines[29]. SE activity is associated with *KLF6* expression in all OC histotypes, but the SE is only mutated in HGSOC (*P* = $8.2 \times 10^{-8}$). *KLF6* is a Krüppel-like transcription factor with tumor suppressor functions, and is associated with chemoresponse and prognosis in OC patients[30,31].

At the 6p22.1 locus, we identified a cluster of seven SNVs in an enhancer located ~9 kb centromeric to the *HIST1* gene cluster (Fig. 3f). The activity of this putative enhancer correlates strongly with the expression of *ZSCAN16* (Spearman's rho = 0.69, *P* = $6.6 \times 10^{-4}$) and *ZSCAN12* (Spearman's rho = 0.47, *P* = 0.02) (Fig. 3g). We used CRISPR/Cas9 to knock out 635 bp of this enhancer in two HGSOC cell lines (UWB1.289 and SHIN3) (Supplementary Fig. 6). Enhancer knockout induced downregulation of both *ZSCAN16* and *ZSCAN12*, and *ZKSCAN3* and *HIST1H2AI*, which are previously predicted targets of this enhancer[32] (Fig. 3h, i and Supplementary Fig. 6). Crucially, there was no change in expression of *ZSCAN31*, a gene not predicted by any method to be targeted by this enhancer. *ZSCAN16, ZSCAN12*, and *ZKSCAN3* are TFs containing SCAN domains that mediate protein–protein interactions; little is known about their function and they have not previously been implicated in OC.

Using motifBreakR[33] we predicted that a somatic SNV in the 6p22.1 enhancer (chr6:27870735:T:A) breaks a TEAD4 motif; this transition was identified in two independent tumors (Fig. 3f). Using publicly available TEAD4 ChIP-seq data in cancer cell lines, we found TEAD4-binding sites overlap 6/8 (75%) FMREs identified in HGSOCs, including the 6p22.1 enhancer. Active TEAD4-binding sites (coinciding with H3K27ac ChIP-seq OC peaks) were significantly mutated in ovarian tumors (fold change (FC)(obs/exp) = 1.8, *P* = $6 \times 10^{-11}$) (Fig. 3j). Globally, we observed that mutations in TEAD4-binding sites occur preferentially in the active TEAD4-binding sites; 196 of 10,872 TEAD4-binding sites (1.8%) and 185 of 1968 active-binding sites (9.4%) harbored SNVs in ovarian tumors. TEAD4 is a known binding partner of PAX8[34,35]. Consistent with this, we found enrichment of SNVs in HGSOCs in active PAX8-binding sites (FC = 1.9, *P* = $2 \times 10^{-10}$). One hundred and nine of 169 HGSOCs (64.5%) harbored a somatic SNV in at least one active PAX8-binding site. Taken together these data indicate that somatic point mutations converge on TEAD4/PAX8-binding sites to deregulate PAX8 target gene expression during OC progression.

**Gene-centric analysis of mutated REs.** We tested the aggregate of somatic mutations in 6,197 CREAGs identified in primary ovarian

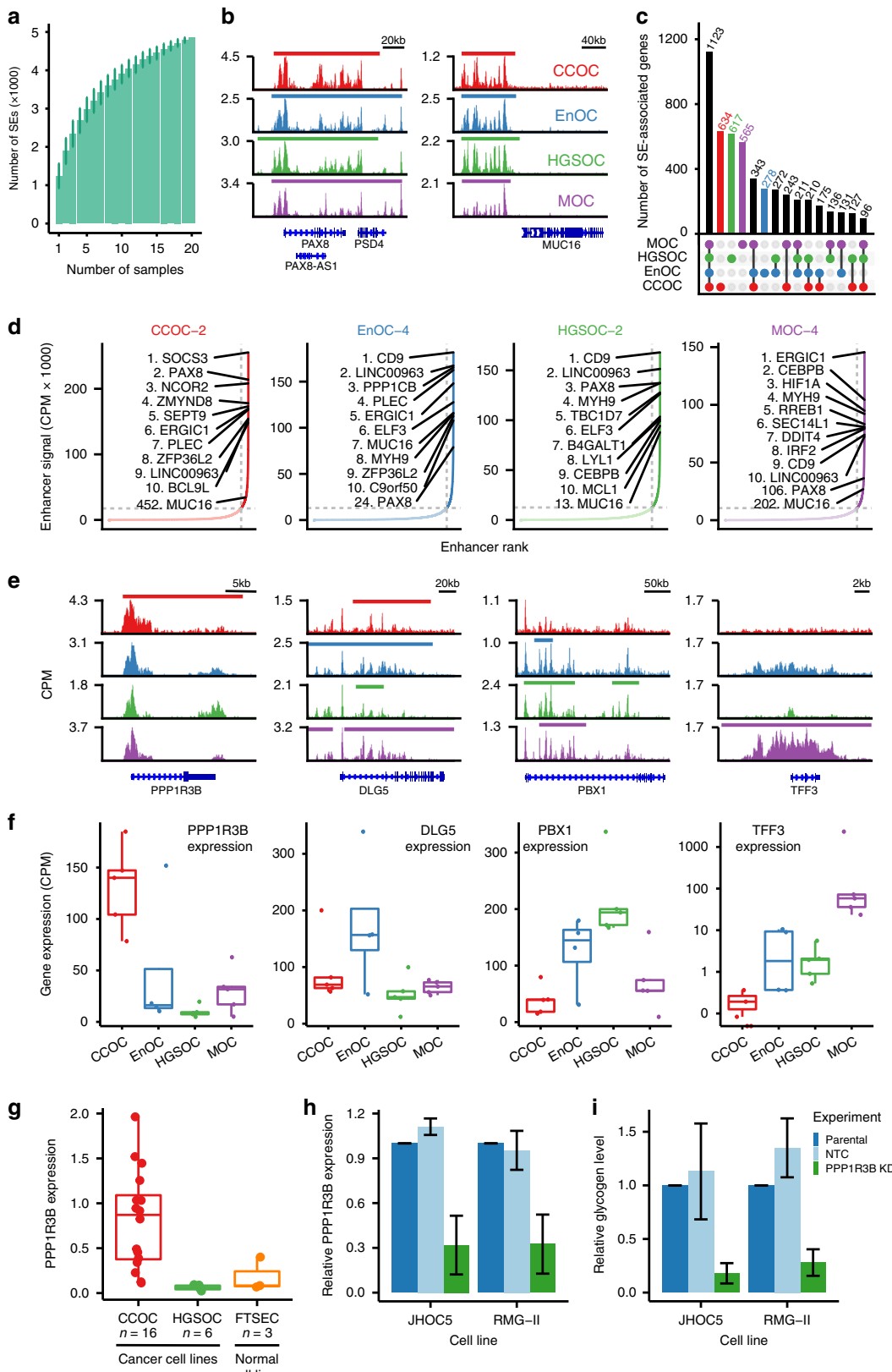

tumors. A total of 916 genes had a significantly increased mutation burden across multiple REs associated with each gene ($P < 0.05$, Fig. 4a–c). There was statistically significant enrichment of SE-associated genes among the most frequently mutated CREAGs ($P = 0.01$, Fig. 4d) which included genes known to be involved in OC pathogenesis; for example hepatocyte nuclear factor 4, gamma

(HNF4G), homeobox D9 (HOXD9)[36], N-myc downstream regulated 1 (NDRG1)[37], CD47 molecule (CD47)[38], MDS1 and EVI1 complex locus (MECOM)[39] and PAX8[29].

We measured differential gene expression as a function of mutational state for all associated REs for each gene. Comparing samples with mutated CREAGs to samples with wild-type

**Fig. 2 Histotype-specific super-enhancers. a** Number of super-enhancers as a function of number of samples. **b** *PAX8* and *MUC16* loci show common OC SEs. **c** UpsetR plot showing the size of the all the subsets of super-enhancer associated genes by whether they are present in CCOC, EnOC, HGSOC, and MOC. **d** Plots of all enhancers ranked by enhancer signal for four representative OC samples, one for each histotype, showing the top 10 SEAGs, and the rank of *PAX8* SE, *MUC16* SE, and *EPCAM* SE for each sample. **e** and **f** Examples of histotype-specific SEAGs. Gene tracks **e** and gene expression **f** that show one example of histotype-specific SEAGs for each histotype (*PPP1R3B*, *DLG5*, *PBX1*, and *TFF3* for CCOC, EnOC, HGSOC, and MOC, respectively). **g** *PPP1R3B* expression in CCOC, HGSOC, and FTSEC cell lines. Relative *PPP1R3B* expression **h** and glycogen level **i** in JHOC5 and RMG-II cell lines before and after *PPP1R3B* knockdown, error bars indicate one standard deviation of the mean values from three independent experiments (performed with technical replicates).

CREAGs, we found 30 significantly differentially expressed genes (Mann–Whitney $U$ test $P < 0.05$). More of these genes were overexpressed ($n = 20$) than down-regulated ($n = 10$) in mutant samples, suggesting gain of function alterations in RE activity are more common than loss of function. Overexpressed genes include prothymosin-alpha (*PTMA*) (FC = 1.26, $P = 0.005$, Mann–Whitney $U$ test); integrator complex subunit 1 (*INTS1*) (FC = 1.44, $P = 0.01$, Mann–Whitney $U$ test); and NOC2 like nucleolar-associated transcriptional repressor (*NOC2L*) (FC = 1.66, $P = 0.01$, Mann–Whitney $U$ test). Analysis of gene essentiality in 517 cancer cell lines (including 53 OC cell lines) show that *PTMA*, *INTS1*, and *NOC2L* are all common essential genes in cancer[40]. Downregulated genes included lysophosphatidic acid receptor 3 (*LPAR3*) (FC = 0.09, $P = 0.007$, Mann–Whitney $U$ test) and metallothionein 1X (*MT1X*) (FC = 0.4, $P = 0.013$, Mann–Whitney $U$ test) (Fig. 4e).

Based on the normalized mutational burden ($P$-value) and the frequency with which somatic SNVs occur in any of the REs that contribute to a CREAG, the *CD47* CREAG (containing five REs) is the most significantly mutated in OC and contains 22 mutations in 21 patients ($P = 6.46 \times 10^{-6}$, Fig. 4c). Overall, the *PAX8* CREAG was the most frequently mutated in HGSOC; 36/169 tumors (21%) harbored a somatic mutation in at least one of three REs in this CREAG ($P = 0.003$) (Fig. 4f). SNVs in the *PAX8* CREAG are scattered throughout PAX8 SE, a 82 kb region overlapping the *PAX8* gene locus (Fig. 4g). The PAX8 CREAG is mutated in other cancer types, but this only reached statistical significance in melanoma, kidney, and OCs (Fig. 4h). Kidney and OCs have high levels of PAX8 expression and cell viability dependency shown by CRISPR screens in cancer cell lines[40], while skin cancer does not show a clear relationship with PAX8 dysregulation. Recent studies have shown that skin cancer has abundant number of mutations in non-coding regions and TF-binding sites, but estimate that a large number of those are random events[41]. Combining these data, we find that 90% of OCs harbor an alteration in the PAX8 pathway, either by somatic amplification (23% of cases) or deletion (13%) of the PAX8 locus, mutation of enhancers upstream of *PAX8* (25%), somatic mutation in PAX8-binding site (61%) or TEAD4-binding site mutation (61%) (Fig. 4i). PAX8 target genes were among the differentially expressed genes associated with CREAG mutational state (Supplementary Fig. 7a) including downregulated expression of *HOXA10*, a gene implicated in the development of endometrioid but not high-grade serous tumors[42] and upregulated expression of *TMPRSS3*, a gene associated with tumorigenic phenotypes in in vitro models of OC[43].

## Discussion

Different cancers have been defined by the spectrum of protein-coding mutations and target genes that drive disease pathogenesis; but little is known about the functional role of non-coding somatic mutations in cancer development which likely drive the underlying mechanisms of gene regulation through epigenomic perturbation. This study describes the histotype-specific architecture of gene regulation in OC based on H3K27ac ChIP-seq analysis of primary tumors representing the four major subtypes of invasive disease. As anticipated, different histotypes of OCs share common epigenomic features, which likely reflects a shared embryologic lineage; but we show conclusively that each subtype also has a unique signature of active enhancers that underpins the histotype-specific patterns of gene expression. Endometrioid OCs (EnOCs) represent an exception in that H3K27ac ChIP-seq analysis only identified a small number of histotype-specific REs. When we focused on the histotype-defining enhancers, two of our EnOCs resembled HGSOC, while the other three resembled CCOC. This is consistent with clinical and biological traits of these tumors; both EnOCs and CCOCs are associated with endometriosis; but late-stage EnOCs can share somatic features with high-grade serous ovarian OCs. This may partly explain the lack of specificity in defining the regulatory landscape of this histotype.

Histotype-specific REs were strongly enriched for putative enhancers. Enhancer depletion was more common than enhancer gain, consistent with previous reports that loss of activity drives cell-type-specific identity[44]. A more in-depth analysis of enhancer activity identified around 1,100 'Müllerian' SEs that are common across all OC histotypes. SEs mark genes associated with cell lineage and cell state, in normal and cancer tissues alike[45]. We identified SEs coinciding with genes that encode established biomarkers in OC, highlighting the disease-specific nature of these findings. This included SEs associated with mucin 16 (*MUC16*) that encodes CA125, and *PAX8*. *MUC16*/CA125 is a serum marker that is used clinically to aid in the diagnosis of OC and monitor disease progression. *PAX8* is a lineage-specific transcription factor that is highly expressed in fallopian tube epithelia, a precursor of HGSOC and is commonly amplified in HGSOCs, emphasizing its essentiality in OC development[8,26]. The *PAX8* SE was detected in all OC histotypes and was most active in HGSOC and EnOC. A SE proximal to *PPP1R3B* was unique to CCOC. *PPP1R3B* was most highly expressed in CCOC compared to other histotypes and knockout of *PPP1R3B* in CCOC cell lines indicates that these analyses have identified functionally relevant biomarkers associated with disease development. We found other histotype-specific SEs which likely regulate genes important for establishing the defining features of each OC histotype that will warrant functional validation in the future. Overall, the data indicate a role for SE landscapes as underlying drivers of histotype-specific OC pathogenesis.

Integrating mutation data from WGS of 232 primary ovarian tumors with epigenomics landscapes, we identified somatic mutations that fall into REs that are candidate non-coding functional targets of these mutations. Ovarian tumors contain several thousand somatic mutations, the vast majority of which (98.7%) lie in the non-protein coding DNA regions; but only a proportion of these are likely to have a functional impact on disease development. There are several hypotheses for the functional mechanisms of non-coding mutations in disease etiology: (1) that somatic mutations perturb specific REs, including active enhancers and SEs, to affect *in cis* the expression of gene(s) that are critical in disease biology; (2) that the constellation of somatic mutations across multiple REs targeting the same gene have equivalent functional effects; and (3) somatic mutations contribute to disease development by impacting either the expression

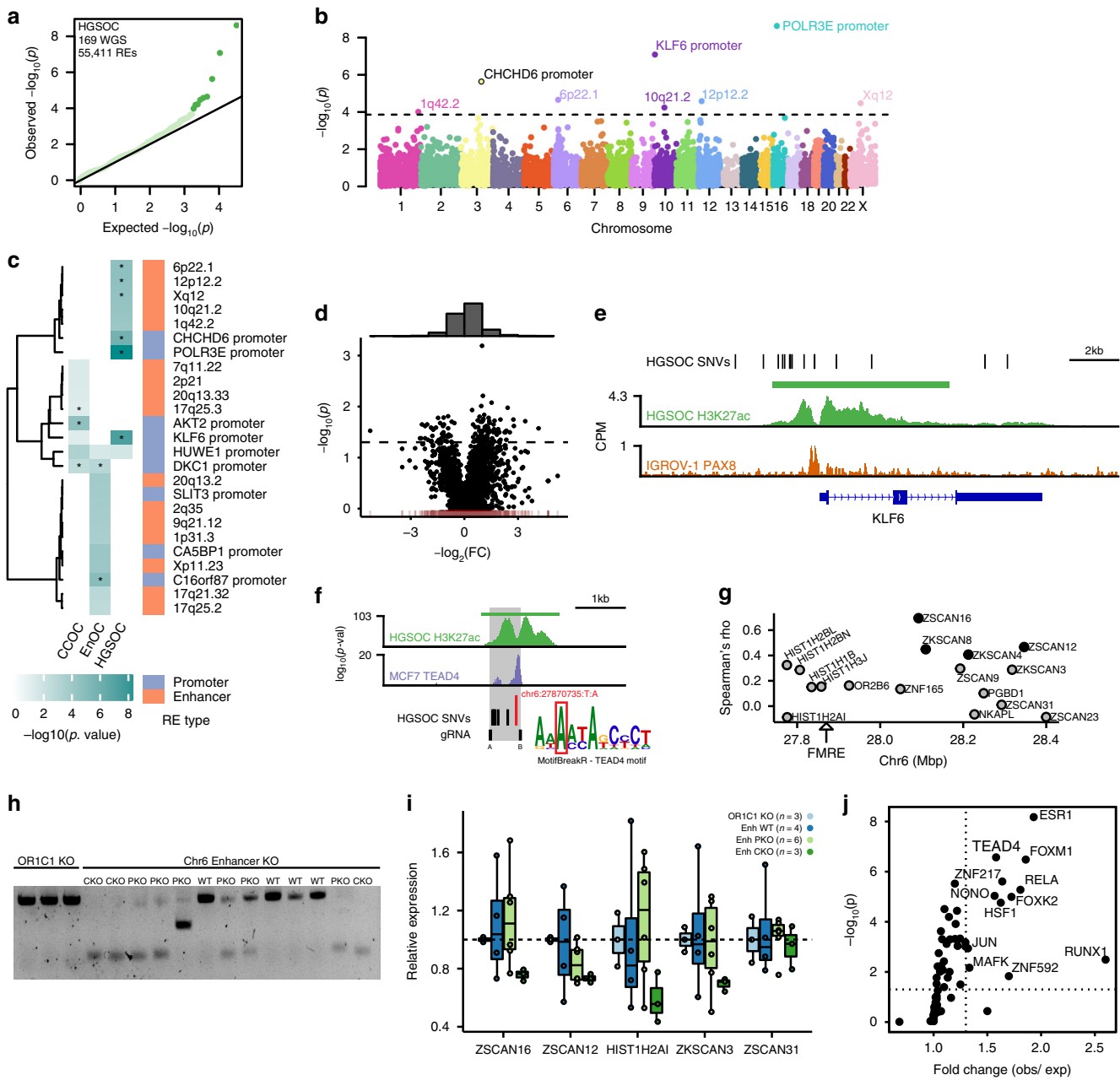

**Fig. 3 Frequently mutated regulatory elements (FMREs) in OC. a** QQ-plot that shows the expected (*x*-axis) and observed (*y*-axis) significance values of the mutational burden for all active REs in HGSOC. **b** Manhattan plot that shows the genomic location (*x*-axis) and significance value (*y*-axis) of the mutational burden for all active REs in HGSOC. **c** Heatmap that shows the mutational burden (*P*-value) of the 25 FMREs across CCOC, EnOC, and HGSOC; asterisks represent FDR ≤ 10%. **d** Volcano plot that shows the fold change of median gene expression (*x*-axis) and the significance value (*y*-axis) of the putative target gene of samples with overlapping single nucleotide variants in an active RE vs. wild-type samples. The histogram on top of the scatterplot shows more overexpression events ($-\log2(FC) > 0$) in the presence of single nucleotide variants than under expression events ($-\log2(FC) < 0$). **e** The *KLF6* locus, location of SNVs, SE, and PAX8-binding sites. **f** The 6p22.1 mutated enhancer, location of SNVs, TEAD4-binding sites and motif logo relative shows position of the recurrent mutation **g** Spearman's rho correlation between the activity of a FMRE (6p22.1 enhancer) and nearby genes. **h** and **i** Single-cell-derived clones after CRISPR/Cas9-mediated deletion in the SHIN3 HGSOC cell line. **h** Gel electrophoresis showing the genotype of the 16 clones (3 *OR1C1* control knockouts (KO), 4 wild type (WT), 6 partial KO (PKO), and 3 complete KO (CKO)). **i** Relative expression of *ZSCAN16*, *ZSCAN12*, *HIST1H2AI*, and *ZSCAN31* in the 16 clones. **j** Fold change and *P*-value of the enrichment of HGSOC somatic SNVs in active TF-binding sites using publicly available MCF-7 TF ChIP-seq data.

or activity of transcription factors (e.g. by altering TF-binding sites). Our analyses found several non-coding REs that contained clusters of somatic mutations (FMREs). FMREs showed histotype-specificity, consistent with the histotype specificity we observed from ChIP-seq and RNA-seq analysis, and were

associated with genes known to be biologically important in OC development. The burden of somatic mutations in REs were also predicted to deregulate pathways that are critical in OC biology. Thus, taken together, our studies support the hypothesis that non-coding somatic mutations within tissue-specific REs perturb

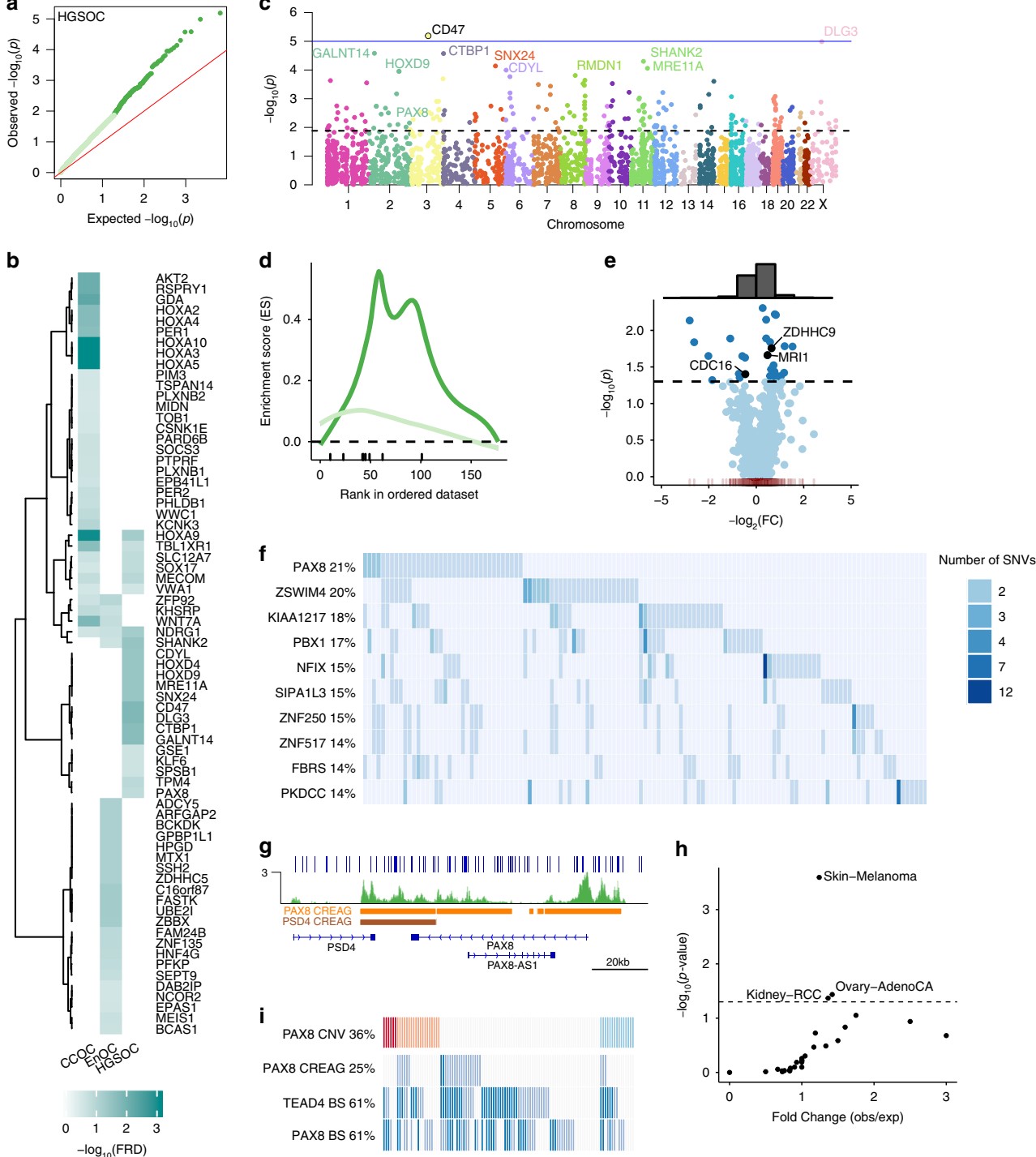

**Fig. 4 Gene-centric mutation rate aggregated across the collection of REs associated to a gene (CREAG). a** QQ-plot that shows the expected (*x*-axis) and observed (*y*-axis) significance values of mutational burden for all CREAGs. **b** Super-enhancer associated genes that have somatic SNVs. **c** Manhattan plot showing the genomic location (*x*-axis) and significance (*y*-axis) of mutational burden for all CREAGs. **d** Enrichment of super-enhancer associated genes in FM CREAGs (dark green) vs. random selection (light green) ($P_{GSEA} = 0.01$). **e** Volcano plot showing the fold change of median gene expression (*x*-axis) and the significance value (*y*-axis) of the putative target gene in samples with single nucleotide variants in a CREAG vs. wild-type samples. The histogram on top of the scatterplot shows more overexpression events ($-\log2(FC) > 0$) in the presence of single nucleotide variants than under expression events ($-\log2(FC) < 0$). **f** HGSOC ($n = 169$) non-coding oncoplot showing the top 10 ranking genes in terms of number of samples with mutations overlapping its CREAG. **g** *PAX8* locus showing the mutations within the *PAX8* CREAG. **h** The number of samples with mutations overlapping the *PAX8* CREAG are statistically significant in Skin-Melanoma, Ovary-Adeno-CA and Kidney-RCC datasets. **i** HGSOC ($n = 110$) oncoplot with *PAX8* copy number variation, *PAX8* CREAG mutation, TEAD4 and *PAX8*-binding site mutations.

regulatory networks that are specifically associated with disease etiology.

These analyses were based on WGS data generated for a relatively small number of primary OCs and for several histotypes. Inevitably, these studies will benefit in the future from additional WGS analyses performed in ovarian tumors for the different histotypes. The greater number of FMREs in endometriosis-associated OCs compared to HGSOCs, despite the smaller sample size, suggests somatic noncoding mutations may play a greater role in the development of clear cell and endometrioid tumors, however, further analysis is needed to properly evaluate this result, since different somatic mutation calling pipelines were used in the generation of this dataset. There are also likely to be limitations in the current analyses because not all SNVs in a RE are likely to have a similar functional impact. Neither did we evaluate other somatic events in REs (e.g. deletions, amplification, and other structural rearrangements) that may impact gene regulation by affecting RE activity.

Perhaps the most compelling finding from these studies is the convergence of several analyses on PAX8 as a major transcription factor target in the etiology of high-grade serous OC. There was a significant clustering of mutations in enhancers upstream of PAX8 in HGSOC but not in CCOC or EnOC; and PAX8-bound active enhancers were also significantly mutated in HGSOC, as were active enhancers bound by TEAD4, a known PAX8-binding partner[29,34]. When we look at the PAX8 CREAG specifically, we do not see evidence for differential expression associated with enhancer/promoter mutations (Supplementary Fig. 7b), but this could be because of our small number of mutated samples with RNA-seq data ($n = 21$), passenger mutations, or heterogeneity of mutation effects. Together these data indicate a role for PAX8 as both a target and mediator of non-coding somatic mutations. This was further supported by the finding of somatic mutations disrupting the TEAD4 motif within a mutated enhancer on chromosome 6. Knockout of the frequently mutated enhancer at 6p22.1 locus validated a positive association between the activity of the enhancer and four genes predicted to be regulated by the enhancer. Further analysis is needed to determine whether there is direct contact between the RE and ZSCAN16, ZSCAN12, HIST1H2AI, and ZKSCAN3 promoters, and to determine the functional role for one or more of these genes in OC pathogenesis. Our studies also identified genes and transcription factors that warrant additional functional studies to validate their role in OC. In addition to PAX8 and TEAD4-bound enhancers, we also identified significantly increased somatic mutation rates in FOXM1 and ESR1 bound regions in HGSOC. Analysis of HGSOC TCGA data indicates FOXM1 is a frequently altered pathway in HGSOC development[8], while ESR1 is expressed in around 80% of HGSOCs[46].

In conclusion, through the integration of tissue-specific epigenomic and gene expression landscapes for the different histotypes of OC with somatic mutation data from WGS analyses of primary tumors, we have identified non-coding elements that may contribute to OC development. Many of the mutated REs and their associated genes provide insights into disease pathogenesis, and the lineage-specific TF PAX8 was identified a central player in the transcriptional dysregulation caused by non-coding somatic mutations in HGSOCs.

## Methods

**Tissue ChIP-seq.** All tissues used were collected with informed consent and the approval of the institutional review boards of the University of Southern California and Cedars-Sinai Medical Center. Tissue ChIP-seq is performed as follows[47]: Briefly, one frozen 3 mm core was pulverized in a pulverization bag (TT05) using the Covaris CryoPrep system (Covaris, Woburn, MA) twice at intensity 4. The tissue was then fixed using 1% formaldehyde (Thermo fisher, Waltham, MA) in Phosphate-buffered saline solution for 10 min at room temperature with rotation

and quenched with 125 mM glycine for 10 min at room temperature with rotation. After rinsing with ice-cold phosphate-buffered saline solution twice, chromatin was resuspended and lysed in ice cold lysis buffer (50 mM Tris, 10 mM EDTA, 1% SDS with protease inhibitor) for 10 min. Chromatin was sheared to 300–500 base pairs using the Covaris E210 sonicator (AFA: 5% duty cycle, 5 intensity, 200 cycles/burst) for 10 min. 1% of chromatin was saved as input for each sample. 5 vol of dilution buffer (1% Triton X-100, 2 mM EDTA, 150 mM NaCl, 20 mM Tris–HCl pH 8.1) was added to the rest of chromatin and the sample was incubated with 1 μg H3K27ac antibody (DiAGenode, C15410196, Denville, NJ; as a ratio of 1:600) coupled with protein A and protein G beads (Life Technologies, Carlsbad, CA) at 4 °C overnight. The chromatin was washed with RIPA washing buffer (0.05 M HEPES pH 7.6, 1 mM EDTA, 0.7% Na deoxycholate, 1% NP-40, 0.5 M LiCl) for five times and, rinsed with TE buffer (pH 8.0) once. The sample was resuspended in elution buffer (50 mM Tris, 10 mM EDTA, 1% SDS), treated with RNase for 30 min at 37 °C, and incubated with proteinase K overnight at 65 °C. Sample DNA and input were extracted using Qiagen Qiaquick columns, and sequencing libraries prepared using the ThruPLEX-FD Prep Kit (Rubicon Genomics, Ann Arbor, MI). Libraries were sequenced using 75-base pair single reads on the Illumina platform (Illumina, San Diego, CA) at the Dana-Farber Cancer Institute.

**ChIP-seq data processing.** The AQUAS pipeline (version 0.3.3)[48] was used to process all H3K27ac ChIP-seq data. Reads were aligned against the reference human genome hg19, filtered by quality and duplication. Several quality control metrics were computed for each individual replicate, including number of reads, percentage of duplicated reads, NSC, RSC, and FRiP (Supplementary Fig. 1). AQUAS follows ENCODE3 guidelines to process ChIP-seq data. For histone modification ChIP-seq data, AQUAS uses macs2 as the peak caller algorithm and a naive overlap approach that selects for the final peak set the regions of the pooled replicate that overlaps 50% or more of each individual replicate. For the 20 OCs, we obtained an average of 33.7 million mapped reads (standard deviation, sd = 9,071,589), and an average of 77,346 peaks (sd = 21,020), per sample. H3K27ac peaks were, on average, 1078 bp wide (sd = 1206 bp), and covered around 83 Mbp per sample (sd = 16 Mbp) (Supplementary Fig. 1).

There was negative correlation between the number of peaks and the average peak width (Pearson's $\rho = -0.58$, $P$-value = 0.007) and positive correlation between number of peaks and genome coverage (Pearson's $\rho = 0.54$, $P$-value = 0.007). Histotype-specific regions were identified using the R Bioconductor package DiffBind[49], which calculates differentially bound regions from multiple ChIP-seq experiments. For each histotype, we selected sites called in at least three out of five samples in the given histotype, with absolute FC value ≥ 3, FDR ≤ 0.05, contrasting the five samples of the histotype of interest against the remaining 15 samples using the method DBA_EDGER. Common regions that were called in all 20 OC H3K27ac ChIP-seq experiments regardless of the intensity of the ChIP-seq signal. The consensus set of H3K27ac ChIP-seq regions for HGSOC were all sites present in at least three (out of five) HGSOC samples, merging overlapping peaks and we recalculated the ChIP-seq score using DiffBind with the new coordinates across all samples, to have homogenous start/end positions for all peaks across the samples.

**RNA-seq.** Primary OC specimens were homogenized, and total RNA was extracted using TRIzol LS (Thermo Fisher Scientific, catalog number: 10296028). Ribosomal RNA (rRNA) was depleted using RiboMinus Transcriptome Isolation Kit (Thermo Fisher Scientific, catalog number: K155002). Poly (A) + RNA was then isolated using Dynabeads Oligo (dT) 25 (Thermo Fisher Scientific, catalog number: 61002). Twenty nanograms rRNA-poly (A) + RNA was used to prepare each RNA-Seq library. External RNA Controls Consortium (ERCC) spike-ins (Thermo Fisher Scientific, catalog number: 4456740) were added as control for normalization of the samples. Strand-specific RNA-Seq libraries were constructed using the NEBNext Ultra Directional RNA Library Prep Kit (NEB, catalog number: E7420). The resulting library concentrations were quantified using the Nanodrop. Libraries were sequenced to generate paired-end 75 bp reads on NextSeq 500 platform (Illumina) in high output running mode. Sequencing was performed at the Molecular Genomics Core facility at the University of Southern California.

**RNA-seq data analysis.** Reads were aligned to hg38 (ref_genome_hg38_gencodev26) using STAR (STAR-2.5.1b), then a read count matrix was generated using featureCounts (version 1.5.0-p1; gencode.v24.annotation.gtf). The samples range from 8.6 to 32.2 million uniquely mapped reads. Histotype-specifically expressed genes were identified using the R Bioconductor package DESeq2 (version 1.24.0) with absolute log2 FC ≥ 2, $P$-value ≤ 0.05, contrasting the five samples (four in case of EnOC) of one histotype against the remaining 14 (15 in case of EnOC).

**RE/gene mapping.** We were unable to collect sufficient tumor material for one of the endometrioid tumors for RNA-seq, therefore, we only have 19 samples (out of 20) with paired ChIP-seq/RNA-seq. We calculated the pairwise Spearman's rho correlation ($\rho$) of the H3K27ac ChIP-seq score of a given RE against the gene expression, in counts per million (CPM), of all genes within the same TAD. We selected RE/gene pairs with $\rho > 0.4$ and one-sided $P$-value < 0.05.

Pathway enrichment analysis is performed using Metascape [https://metascape.org] using as input the list of genes associated with CCOC-specific, HGSOC-

specific, and MOC-specific (enriched and depleted separately) REs against KEGG pathway.

**Filtering SNVs**. We used SNVs from 232 OCs (110 from the PCAWG cohort and 122 from the UBC cohort). We removed all SNVs that overlap coding regions, regions annotated as low mappability regions by ENCODE (wgEncodeDacMappabilityConsensusExcludable.bed), as well as areas of the genome that are prone to cause false positives in ChIP-seq assays (seq.cov1.ONHG19.bed.gz). For HGSOC ($n = 169$), there is a total of 1,276,929 SNVs that fall into 1,276,108 individual positions (1,275,309 positions with only 1 mutated sample, 784 positions with two mutated samples, 11 positions with three mutated samples, three positions with four mutated samples, and one position (chr3:174499208-174499208) with seven mutated samples).

**Identifying FMREs**. To identify FMREs we used a Poisson binomial distribution (PBD) with a vector of probabilities, i.e., one background mutation rate for each sample. Let $X_i(X_i \in [0, 232])$ be a random variable that represents the number of samples with at least one mutation in the $i$th RE, then $X_i$ follows a PBD with a vector of probabilities $p = [1 - (1 - p_k)^{n_i}]_k$, where $n_i$ is the size of the $i$th RE in base pairs, and $p_k$ is the global background rate of sample $k$ ($k \in [1, 232]$) empirically estimated by the ratio of the total number of non-coding SNVs in sample $k$ ($n_k$) over the total coverage of the regions of interest in base pairs, (the H3K27ac-positive regions) ($n_{cov}$):

$$p_k = \frac{n_k}{n_{cov}}$$

To determine whether the observed number of mutated samples in the $i$th RE ($s_i$), we calculate the probability of having at least $s_i$ samples mutated, i.e., $P\text{-value}_i = P(X_i \geq s_i)$. $P$-values were adjusted used the Benjamini–Hochberg method. For gene-level analysis, we combined all REs and annotated promoters associated to gene $j$ into a pseudo-RE and estimated the probability of having $s_j$ or more mutated samples in all combined REs, i.e., $P\text{-value}_j = P(X_j \geq s_j)$, where $X_j$ follows a PBD with a vector of probabilities $p = [1 - (1 - p_k)^{n_j}]_k$, where $n_j$ is the total length of the pseudo-RE, i.e., the sum of the widths of all REs and annotated promoters associated to gene $j$.

**Gene expression changes associated with RE mutations**. For 89 WGS samples (out of 110) in PCAWG, RNA-Seq was available. Analysis was restricted to genes with at least two mutated samples in the RE overlapping the promoter or an associated RE. For each gene, we calculated the FC, defined as the ratio of the median gene expression of the gene in the mutated samples ($x_1$) over the median gene expression of the gene in the non-mutated samples ($x_2$).

**Enhancer deletions in OC cell lines**. CRISPR/Cas9 system was used to delete the chr6 enhancer in OC cells. FUCas9Cherry (Gifts from Dr. Marco Herold, Addgene plasmid number 70182) was transfected together with lentiviral packaging plasmids pMD2.G and psPAX2 (Gifts from Dr. Didier Trono, Addgene plasmid numbers 12259 and 12260) into HEK293T cells. UWB1.289 and SHIN3 cells were then transduced with lentivirus and mCherry-positive cells (UWB1.289/Cas9 and SHIN3/Cas9) were sorted by FACS. Cas9 activity was confirmed using gRNAs targeting the *RB1* locus, and Surveyor T7 Endonuclease 1 digests. UWB1.289/Cas9 and SHIN3/Cas9 were subsequently transduced by lentivirus containing either gRNA pair targeting enhancer region on chromosome 6 (Chr6) or gRNA pair targeting a control gene *OR1C1* before selection with 400 ng/mL puromycin. Plasmids expressing gRNAs were obtained from Transomic Technology. Sequences for Chr6_gRNA-A are: TCCCTTGCCAGCTCACTCAA; Chr6_gRNA-B: AGGAATCCAACTAATACCAT; OR1C1_gRNA-A: AGGGCTGAAATAGACGGCGA; OR1C1_gRNA-B: AGAGGTGATCTTCAGAACAG. Progeny of edited cells were sorted by FACS into single cells, based on mCherry expression. The following primers were used for PCR to validate the corresponding genome type after editing: Chr6-F: TGCTTCCTGATTTTCTCCTCA; Chr6-R: CCTGAAAAGAAGGGAAGAAGG; OR1C1-F: GCATTTTCTGAAGTCCCCTCT; OR1C1-R: TGTGCCTCCAATATCATCCA.

**Gene expression measurement by RT-qPCR**. Total RNA was extracted from bulk cells or single cell colonies with either knockout (KO) of the targeted enhancer on Chr6 or control gene *OR1C1* using the NucleoSpin RNA extraction kit (Macherey-Nagel, Catalog number: 740984.250). Total RNA was reverse transcribed into cDNA using M-MLV reverse transcriptase (Promega, Catalog number: M1705) according to the manufacturer's instructions. Targeted gene expression analysis was performed using KAPA SYBR FAST (Millipore Sigma, Catalog number: KK4618)-based PCR using the following primers: ZSCAN16-F: CTCCTCAGCATCCTAAGTCCAAA; ZSCAN16-R: GCTATGACTGAAACTTTTCCCACAT; ZSCAN12-F: GCAGAGAGGTCTTCCGTCAG; ZSCAN12-R: AGACCTGCTCTCCTGGTTCA; ZKSCAN3-F: GAGCTTCCAGAAAAGGAGCAT; ZKSCAN3-R: CTTTCCACATTCATGGCAGAT; HIST1H2AI-F: AACGATGAGGAGCTCAACAAG; HIST1H2AI-R: GCTCTGAAAAGAGCCTTTGGT; ZSCAN31-F: CTGAAGTGCTCTTGGAGGATG; ZSCAN31-R: CCCCAAATGTTCCATTTCTTT; U6-F: GCTTCGGCAGCACATATA

CTAAAAT; U6-R: CGCTTCACGAATTTGCGTGTCAT.U6 was used for normalization.

**Reporting summary**. Further information on research design is available in the Nature Research Reporting Summary linked to this article.

## Data availability
The RNA-sequencing and ChIP-sequencing data have been deposited in the GEO database under the accession code GSE121103. The whole-genome sequencing data is available from the PCAWG database under the search "Primary Site=Ovary" and "Software=PCAWG SNV-MNV callers" [https://dcc.icgc.org/pcawg]. All the other data supporting the findings of this study are available within the article and its supplementary information files and from the corresponding author upon reasonable request. A reporting summary for this article is available as a Supplementary Information file.

## Code availability
Custom code is available at the Lawrenson Lab GitHub repository [https://github.com/lawrenson-lab].

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

## Acknowledgements

This work was supported by a Tower Cancer Research Foundation Career Development Award and a K99/R00 grant from the National Cancer Institute (NCI) (1K99CA184415-01), both to K.L. K.L. is supported in part by a Liz Tilberis Award from the Ovarian Cancer Research Alliance (OCRA, Grant number 599175). The tissue specimens were either collected as part of the USC Jean Richardson Gynecologic Tissue and Fluid Repository, which is supported by a grant from the USC Department of Obstetrics & Gynecology and the NCT Cancer Center Shared Grant award P30 CA014089 (to the Norris Comprehensive Cancer Center) or as part of the Women's Cancer Biobank at Cedars-Sinai Medical Center. This work was supported in part by the Ovarian Cancer Research Fund Alliance Program Project Development Grant (373356): Co-Evolution of Epithelial Ovarian Cancer and Tumor Stroma (B.Y.K., K.L.). Part of these data were from the PanCancer Analysis of Whole Genomes (PCAWG) project [https://dcc.icgc.org/pcawg].

## Author contributions

Conception and design of the work (R.I.C., S.A.G., M.L.F., K.L.); provided tissue specimens (Y.G.L., B.Y.K.); pathology review of tumors (P.Y.M.-F.); performed the ChIP-seq (J.-H.S.); performed the RNA-seq and genome editing (X.L.); performed the *PPP1R3B* KD (F.A.); provided somatic variant data (S.P.S., D.G.H., B.P.B.); performed most of the analysis of data (R.I.C.); additional analysis (D.J.H., J.R., M.A.S.F.); interpretation of the data (R.I.C., A.G., B.P.B., S.A.G., M.L.F., K.L.). Wrote the paper (R.I.C., S.A.G., M.L.F., K.L.). All authors read and approved the final manuscript.

## Competing interests

The authors declare no competing interests.
