## [Peer Review File · Nature Communications]

Reviewers' comments:

Reviewer #1 (Remarks to the Author): Expertise in ovarian cancer genomics

This manuscript addresses an important genomic question in ovarian cancer: namely, what are the functional consequences of mutations in the non-coding genome? This manuscript represents the first detailed genomic survey combining ChIP-seq data with mutation and gene expression analysis.

The majority of the data is observational based on a detailed study of 20 ovarian cancers with ChIP-seq and RNAseq profiling and downstream correlative analysis with an independent set of 232 ovarian cancers using whole genome sequencing data and in a subset, RNA expression data.

Main comments

1. The information presented in the text of the manuscript and figures is extremely detailed and dense. Use of visual abstracts and stronger signposting through the MS would make the messages easier to understand for the general reader. Some of the figures need improved labelling—for example, in Fig 1 consistent colour coding is used to indicate the ovarian cancer histotype but this is not highlighted in the figure legend or on individual sub figures.

2. The clinical data for the 20 primary tumours described in Fig 1 is limited and it is unclear what quality assurance or other mutational profiling has been performed on the samples. This is important as in the discussion (but not in the results) the possibility of misdiagnosis for two of the endometrioid cases is presented. Did other genomic data from these cases indicate diagnostic uncertainty? Was histological review performed on these cases? Could this have skewed downstream analyses?

3. The discussion addresses limitations for the ChIP-seq and downstream data analysis, particularly that there has been no evaluation of the effects of other somatic events. However, would high levels of copy number amplification skew the analyses by over representing amplified regions of the genome in both sequencing and ChIP experiments? Would this impact on the analysis looking for common active regulatory elements and super enhancers?

4. The data presented on super enhancer landscapes is persuasive and nicely correlated by proof of principle functional studies on protein phosphatase 1 regulatory subunit 3B.

5. I am less convinced by the comparative analysis of mutations in regulatory elements and super enhancers in the independent ovarian cancer cohort. The identification of frequently mutated regulatory elements was carried out at the false discovery rate of 0.25 (25% of results are expected to be false positive). Eight cases with mutations were identified from high-grade serous ovarian carcinoma cases and 17 were identified across the endometrioid and clear cell ovarian cases. Given the overrepresentation of high-grade serous ovarian cases in the group how do the authors account for this imbalance? Although the authors assert that 89 mutation associated genes were significantly overexpressed, it is not clear whether any of the mutations alter PAX8 transcription or PAX8 protein function.

6. In the gene centric analysis of frequently mutated regulatory elements, a mutation rate of 21% was found for the PAX8 CREAG but this extends over a large domain of 100kb. This data needs more explanation, particularly to reassure the reader that this is not false discovery. The authors state that they have controlled these comparisons based on "the normalized mutational burden and the frequency with which somatic SNVs occur in any of the REs that contribute to a CREAG". What exactly was done? Does the background mutation rate in RE correlate with the level of gene expression and the degree of open chromatin (length, accessibility) and was this controlled for in comparisons? Could this explain apparent increased substitution mutations in PAX8 regulatory

elements as this is a highly expressed domain in HGSOc cases? The finding that melanoma has significant mutational burden in PAX8 RE is also concerning—is further correction required? The summary sentence suggesting that 90% of OC harbour an alteration in the PAX8 pathway is not very precise. What were the criteria for amplification and how many of the cases had high level amplification of PAX8? Although disparate processes may affect the PAX8 locus, the data presented does not demonstrate consistent up-regulation of PAX8 function and the MS does not present any rationale as to why loss of PAX8 function would be beneficial to HGSOc cells. Could the majority of the somatic mutations identified be passenger mutations?

James Brenton

Reviewer #2 (Remarks to the Author): Expertise in ovarian cancer epigenetics

The authors provide a computational rich study that describes new data on non-coding mutations in ovarian cancer and frequently mutated regulatory elements or FMREs in the disease. Data bases as well as Chip-seq (H3K27ac), RNA-seq and whole-genome-seq data were used in an integrated fashion. Functional validation of some of the findings was done using CRISPR/Cas9 to knockout an enhancer and examine the effect on gene expression. The findings on PAX8 transcriptional network are novel and particularly relevant to ovarian cancer etiology and biology. Overall this is an exciting study from an expert group of ovarian cancer investigators that goes well beyond the current state of genomic alterations in the field and opens up new areas of investigation for ovarian and other cancers.

A concern is that the key analysis used only 5 samples per group and grade/stage for some of the histotypes are early and for others only late stage samples were included. EnOC included early and late stages- mixed group. Did stage have any effect on the various landscape analyses? It seems that early late stages could alter the somatic landscapes. Furthermore, of the 20 samples, 19 were used for RNA-seq, but more explanation as to why one of the samples was not available for transcriptomic analysis is needed.

Other comments include

FMREs- A false discovery rate (FDR) of 25% (FDR 0.25) for the 25 FMREs across all histotypes does not seem stringent (Figures 3a-c) to draw conclusions and higher than what is typically used (and for the data in 3c the p. value is shown). MOC histotype is missing from this figure. Check figure 4 legend for text corresponding to g & h

Reviewer #1:

GENERAL COMMENTS: This manuscript addresses an important genomic question in ovarian cancer: namely, what are the functional consequences of mutations in the non-coding genome? This manuscript represents the first detailed genomic survey combining ChIP-seq data with mutation and gene expression analysis.

The majority of the data is observational based on a detailed study of 20 ovarian cancers with ChIP-seq and RNAseq profiling and downstream correlative analysis with an independent set of 232 ovarian cancers using whole genome sequencing data and in a subset, RNA expression data.

COMMENT 1. The information presented in the text of the manuscript and figures is extremely detailed and dense. Use of visual abstracts and stronger signposting through the MS would make the messages easier to understand for the general reader. Some of the figures need improved labelling—for example, in Fig 1 consistent colour coding is used to indicate the ovarian cancer histotype but this is not highlighted in the figure legend or on individual sub figures.

RESPONSE: Thank you for bringing this to our attention. We have now added a visual abstract that shows a summary of the study (new **Figure 1a**). We have also modified the figures and updated the figure legends to have consistent colour coding across the manuscript that relates to the ovarian cancer histological subtypes.

Figure 1a. Study overview - The implementation of the use of landscapes of active chromatin in ovarian cancer to identify frequently mutated regulatory elements.

COMMENT 2. The clinical data for the 20 primary tumours described in Fig 1 is limited and it is unclear what quality assurance or other mutational profiling has been performed on the samples. This is important as in the discussion (but not in the results) the possibility of misdiagnosis for two of the endometrioid cases is presented. Did other genomic data from these cases indicate diagnostic uncertainty? Was histological review performed on these cases? Could this have skewed downstream analyses?

RESPONSE: One of the goals of this study is to identify regulatory elements specifically active (or inactive) in each ovarian cancer histological subtype. Therefore, accurate histological subtype annotation is essential, and so all cases were reviewed by an expert Gynecologic Pathologist with extensive experience in ovarian cancer (PMF). Based on her expert review of the H&E stained slides, we do not think that the endometrioid cases in our study are misdiagnosed. For clarity we have therefore modified the discussion as follows:

“...both EnOCs and CCOCs are associated with endometriosis; but late-stage EnOCs can share somatic features with high-grade serous ovarian OCs (HGSOcs). This may partly explain the lack of specificity in defining the regulatory landscape of this histotype.”

We do agree that coding somatic mutation data could be very informative in these cases; genome profiling data are not currently available for this cohort, although this is something we hope to incorporate in the future.

COMMENT 3. The discussion addresses limitations for the ChIP-seq and downstream data analysis, particularly that there has been no evaluation of the effects of other somatic events. However, would high levels of copy number amplification skew the analyses by over representing amplified regions of the genome in both sequencing and ChIP experiments? Would this impact on the analysis looking for common active regulatory elements and super enhancers?

RESPONSE: Copy number amplification could affect epigenome profiling results, and there are examples of super-enhancers coinciding with regions of copy number amplification^{1,2}. To correct for this, we sequenced input

DNA and used it as part of our peak calling pipeline, which calculates a ChIP-seq score after subtracting the input DNA signal to permit sample-matched correction for copy number variation to be performed³.

COMMENT 4. The data presented on super enhancer landscapes is persuasive and nicely correlated by proof of principle functional studies on protein phosphatase 1 regulatory subunit 3B.

RESPONSE: Thank you!

COMMENT 5. I am less convinced by the comparative analysis of mutations in regulatory elements and super enhancers in the independent ovarian cancer cohort. The identification of frequently mutated regulatory elements was carried out at the false discovery rate of 0.25 (25% of results are expected to be false positive). Eight cases with mutations were identified from high-grade serous ovarian carcinoma cases and 17 were identified across the endometrioid and clear cell ovarian cases. Given the overrepresentation of high-grade serous ovarian cases in the group how do the authors account for this imbalance? Although the authors assert that 89 mutation associated genes were significantly overexpressed, it is not clear whether any of the mutations alter PAX8 transcription or PAX8 protein function.

RESPONSE: We will address each of the concerns raised in this comment individually:

(1) False discovery rate threshold selection - An FDR cutoff of 0.25 indicates that the result is likely to be valid 3 out of 4 times, which is reasonable in the setting of exploratory discovery where one is interested in finding candidate hypothesis to be further validated. Given the relatively small number of samples being analyzed, using a more stringent FDR cutoff may lead to overlook potentially significant results. The Pan-Cancer Analysis of Whole Genomes (PCAWG) consortium reports significant results using $FDR \leq 0.10$ and near-significant results with $0.10 < FDR \leq 0.25$ ⁴, therefore, we updated our results to specify both FDR cutoffs (**Figure 3c**).

(2) Histotype-specific mutated elements - To account for the overrepresentation of high-grade serous ovarian cancer cases in the group, we performed independent analysis for each ovarian cancer histological subtype. As noted, we identified 8 regions specific to HGSOC and 17 for the endometrioid and clear cell cases, despite the smaller sample size. It is important to note that we collected WGS datasets from two different cohorts (110 HGSOC samples from PCAWG and the rest from Wang et al. 2017⁵), which utilized different somatic mutation calling pipelines. To compare fairly the somatic burden between ovarian cancer histologic subtypes (CCOC, EOC and HGSOC), it would be necessary to harmonize the somatic mutation calling for all samples, a task out of the scope of this study. Nonetheless, assuming the somatic mutation datasets are comparable, we checked what are other factors that contribute to the enrichment of frequently mutated regulatory elements in endometriosis associated ovarian cancers, and found that, due to the difference in number of samples with WGS for each cancer type (169 for HGSOC, 35 for CCOC and 28 for EnOC), the number of active regulatory elements with at least one mutation is significantly different for each subtype (14398 for HGSOC, 2225 for CCOC and 4960 for EnOC). These active regulatory elements were used for discovery of frequently mutated regulatory elements, and for multi-hypothesis correction, resulting in a more stringent p-value cutoff for HGSOC (with ~14 thousand independent REs/tests) than for CCOC (with ~ 2 thousand independent REs/tests) or EnOC (with ~5 thousand independent REs/tests). For the reasons presented above, we prefer not to present this result but instead to include more speculative language in the discussion:

"The greater number of FMREs in Endometriosis-associated (EaOCs) compared to HGSOCs, despite the smaller sample size, suggests somatic noncoding mutations may play a greater role in the development of clear cell and endometrioid tumors, however, further analysis is needed to properly evaluate this result, since different somatic mutation calling pipelines were used in the generation of this dataset"

(3) Integrated analyses of gene expression and regulatory element mutations – This response relates to the questions about the impact of mutations on PAX8 function. Given that our analyses excluded exonic mutations, it is unlikely that PAX8 protein function is impacted by these variants, however, the expression of PAX8 could be altered. In addition, if somatic mutations reside within PAX8 binding sites, the ability of PAX8 to bind to the

genome and regulate target gene expression could also be affected. We explored these two avenues using the samples for which we have RNA-seq and WGS data. Because we had only paired RNA-seq and WGS data for 89 high-grade serous tumors, we had limited power to detect associations between somatic mutation and gene expression (this is a common problem in this field). On average only 2.48 tumors (range = 2-13) harbored a mutation in each enhancer of interest in the enhancer-level analyses; in the gene-level analyses the average number of mutations in the ‘mutated’ group was 4.19 (range = 2-24). As we have very few recurrent mutations at the base-pair level, these analyses are based on the assumption that all mutations associated with a gene of interest will have the same effect (i.e., all will be activating, or all will be inactivating), an assumption which is unlikely to always be met. For these reasons, we focused on asking whether we could detect global trends in gene expression associated regulatory elements. We consistently find that noncoding mutations are associated with higher target gene expression compared in wild-type tumors, with regulatory element mutations around twice as likely to be activating than inactivating. When we look at the PAX8 CREAG specifically, we do not see evidence for differential expression associated with enhancer/promoter mutations (**Figure B**), but this could be because of our small number of mutated samples with RNA-seq data (n=21), passenger mutations, or heterogeneity of mutation effects, as described above.

Figure A. PAX8 expression for PAX8 CREAG mutant samples (n=21) and wild-type (n=68).

Based on this reviewer comment, we also examined sets of PAX8 target genes from Adler et al., 2017⁶ and Elias et al., 2016⁷. First, we looked at the targets of REs with significant association between mutational state and gene expression. Among PAX8 targets, we found *TMPRSS3* and *SOX17* (**Figure Ci**) to be significantly associated with mutational state, however, we did not find an enrichment of PAX8 targets among the differentially expressed genes associated with RE mutational state (odds ratio = 0.78, p-value = 0.615, two-sided Fisher’s exact test). Second, we examined genes differentially expressed in our FM CREAG analyses (**Figure Cii**). Some PAX8 target genes were among the differentially expressed genes associated with CREAG mutational state (odds ratio = 1.22, p-value = 0.25, one-sided Fisher’s exact test), including downregulated expression of *HOXA10*, a gene implicated in the development of endometrioid but not high-grade serous tumors⁸ and upregulated expression of *TMPRSS3*, a gene associated with tumorigenic phenotypes in *in vitro* models of ovarian cancer⁹. We have now revised the manuscript and added these details to the main results, as follows:

Figure B. Differential gene expression in the presence/absence of associated RE mutations (i) or CREAG mutations (ii) highlighting significant (p-value < 0.05) PAX8 target genes.

“PAX8 target genes were among the differentially expressed genes associated with CREAG mutational state (**Supplementary Figure S7**) including downregulated expression of *HOXA10*, a gene implicated in the development of endometrioid but not high-grade serous tumors and upregulated expression of *TMPRSS3*, a gene associated with tumorigenic phenotypes in *in vitro* models of ovarian cancer.”

COMMENT 6: In the gene centric analysis of frequently mutated regulatory elements, a mutation rate of 21% was found for the PAX8 CREAG but this extends over a large domain of 100kb. This data needs more explanation, particularly to reassure the reader that this is not false discovery. The authors state that they have controlled these comparisons based on “the normalized mutational burden and the frequency with which somatic

SNVs occur in any of the REs that contribute to a CREAG". What exactly was done? Does the background mutation rate in RE correlate with the level of gene expression and the degree of open chromatin (length, accessibility) and was this controlled for in comparisons? Could this explain apparent increased substitution mutations in PAX8 regulatory elements as this is a highly expressed domain in HGSOV cases? The finding that melanoma has significant mutational burden in PAX8 RE is also concerning—is further correction required? The summary sentence suggesting that 90% of OC harbour an alteration in the PAX8 pathway is not very precise. What were the criteria for amplification and how many of the cases had high level amplification of PAX8? Although disparate processes may affect the PAX8 locus, the data presented does not demonstrate consistent up-regulation of PAX8 function and the MS does not present any rationale as to why loss of PAX8 function would be beneficial to HGSOV cells. Could the majority of the somatic mutations identified be passenger mutations?

RESPONSE: To test for frequently mutated CREAGs we created a ‘pseudo-element’ which contained all the elements associated with each gene. We then modeled the number of mutated samples as a Poisson Binomial Distribution with probability of success $(1-(1-p))^n$, where n is the length of the region of interest, i.e., PAX8 CREAG (82 Kbp long) and p is the mutation rate for each sample. So, the identification of frequently mutated CREAGs is adjusted for CREAG size. Therefore, while it is true that larger frequently mutated CREAGs will be mutated in a greater proportion of patients, our process for identifying frequently mutated CREAGs is not biased by CREAG length.

The significant mutation burden of the PAX8 CREAG in melanoma was unexpected. We note that melanoma is an outlier in that it has a substantially higher global mutation burden than other tumor types in PCAWG. Using the set of 147 melanoma WGS samples from PCAWG, we calculated the somatic burden in all HGSOV CREAGs and observed that melanoma has a wider range of significance scores ($-\log_{10}(p\text{-value})$) when compared to HGSOV (**Figure D**). We adjusted for background mutation rate using the universe of active chromatin in ovarian tumors, which will include active enhancers specific to ovarian tumors and absent in melanoma. Since there is an established link between chromatin activity and mutation rate (mutation rate is lower in active chromatin), it is plausible that this background correction underestimated the background mutation rate in melanoma.

Figure C. Mutational burden of HGSOV CREAGs using HGSOV or skin cutaneous melanoma (SKCM).

As described in our response to comment 5, we were underpowered to detect associations between PAX8 CREAG mutations and PAX8 expression, but functional validation of these results is part of our ongoing research. We have modified the language of the results to use more speculative language about the possible effects of PAX8 CREAG mutations, including the likelihood that a proportion could be passenger events:

“To understand the functional effect of somatic mutations at PAX8 CREAG we would need to be able to discriminate between mutations that promote PAX8 expression, mutations that inhibit PAX8 expression and mutations that have no effect in PAX8 expression, however, with the available data we are not able to predict the effect of these mutations on PAX8 expression and further functional validation is required.”

Reviewer #2:

The authors provide a computational rich study that describes new data on non-coding mutations in ovarian cancer and frequently mutated regulatory elements or FMREs in the disease. Data bases as well as Chip-seq (H3K27ac), RNA-seq and whole-genome-seq data were used in an integrated fashion. Functional validation of some of the findings was done using CRISPR/Cas9 to knockout an enhancer and examine the effect on gene expression. The findings on PAX8 transcriptional network are novel and particularly relevant to ovarian cancer etiology and biology. Overall this is an exciting study from an expert group of ovarian cancer investigators that goes well beyond the current state of genomic alterations in the field and opens up new areas of investigation for ovarian and other cancers.

COMMENT 1: A concern is that the key analysis used only 5 samples per group and grade/stage for some of the histotypes are early and for others only late stage samples were included. EnOC included early and late stages- mixed group. Did stage have any effect on the various landscape analyses? It seems that early late stages could alter the somatic landscapes. Furthermore, of the 20 samples, 19 were used for RNA-seq, but more explanation as to why one of the samples was not available for transcriptomic analysis is needed.

RESPONSE: We wanted to include the four major ovarian cancer histologic subtypes to have a comprehensive landscape of ovarian cancer active regions, but as a result we were limited to 5 samples per histotype. We note that in the field of tissue ChIP-seq, 20 tumors is a relatively large sample set, however, with 5 samples per histologic subtype, we are underpowered to detect small changes between the histologies, or associated with stage in one histotype specifically. We have now included stage as a variable in **Figure 1c**, and noted that lower grade endometrioid tend to be associated with CCOC, whereas the higher grade EnOC resembled HGSOC. Globally we do not see evidence that stage or grade impacts the landscape of active chromatin in this cohort, but reiterate that our study was not designed to detect stage-specific epigenomic features within each histologic subtype.

We were not able to perform RNA-seq on one of the endometrioid tumors as we had insufficient tumor material. We have now added this additional detail to the methods, as follows:

“We were unable to collect sufficient tumor material for one of the endometrioid tumors for RNA-seq, therefore, we only have 19 samples (out of 20) with paired ChIP-seq/RNA-seq.”

COMMENT 2: FMREs- A false discovery rate (FDR) of 25% (FDR 0.25) for the 25 FMREs across all histotypes does not seem stringent (Figures 3a-c) to draw conclusions and higher than what is typically used (and for the data in 3c the p. value is shown). MOC histotype is missing from this figure.

Check figure 4 legend for text corresponding to g & h

RESPONSE: As described in our response to Comment 5 from reviewer 1, the FDR cutoff of 25% is consistent with previous analogous analyses performed in other studies⁴. We elected to be more inclusive with our FDR threshold as we have a relatively small number of samples and a more stringent cutoff would result in overlooking potentially significant results. This has been previously observed in other similar studies¹⁰. We have now modified **Figure 3c** to stratify significant FMREs ($FDR \leq 0.10$) from near-significant results ($0.10 < FDR \leq 0.25$), similar to a study by the Pan-Cancer Analysis of Whole Genomes (PCAWG) consortium⁴.

We gathered a dataset of whole genome sequence data for 232 ovarian cancers, which included 28 clear cell, 35 endometrioid and 169 high-grade serous tumors. There are currently no WGS available for mucinous ovarian cancer and so this histotype was excluded from the histotype-specific mutation analyses in the latter part of the manuscript.

We have now revised Figure 4 legend, as follows:

“...(g) PAX8 locus showing the mutations within the PAX8 CREAG. (h) The number of samples with mutations overlapping the PAX8 CREAG are statistically significant in Skin-Melanoma, Ovary-Adeno-CA and Kidney-RCC datasets. (i) HGSOE (n=110) oncoplot showing PAX8 copy number variation, PAX8 CREAG mutation, TEAD4 and PAX8 binding site mutations.”

Bibliography

1. Zhang, X. *et al.* Somatic Superenhancer Duplications and Hotspot Mutations Lead to Oncogenic Activation of the KLF5 Transcription Factor. *Cancer Discov* **8**, 108–125 (2018).

2. Zhang, X. *et al.* Identification of focally amplified lineage-specific super-enhancers in human epithelial cancers. *Nature Genetics* **48**, 176–182 (2016).
3. Shin, H., Liu, T., Duan, X., Zhang, Y. & Liu, X. S. Computational methodology for ChIP-seq analysis. *Quant Biol* **1**, 54–70 (2013).
4. Reyna, M. A. *et al.* Pathway and network analysis of more than 2,500 whole cancer genomes. *bioRxiv* 385294 (2018) doi:10.1101/385294.
5. Wang, Y. K. *et al.* Genomic consequences of aberrant DNA repair mechanisms stratify ovarian cancer histotypes. *Nature Genetics* **49**, 856 (2017).
6. Adler, E. K. *et al.* The PAX8 cistrome in epithelial ovarian cancer. *Oncotarget* **8**, 108316–108332 (2017).
7. Elias, K. M. *et al.* Epigenetic remodeling regulates transcriptional changes between ovarian cancer and benign precursors. *JCI Insight* **1**, (2016).
8. Cheng, W., Liu, J., Yoshida, H., Rosen, D. & Naora, H. Lineage infidelity of epithelial ovarian cancers is controlled by HOX genes that specify regional identity in the reproductive tract. *Nat Med* **11**, 531–537 (2005).
9. Zhang, D., Qiu, S., Wang, Q. & Zheng, J. TMPRSS3 modulates ovarian cancer cell proliferation, invasion and metastasis. *Oncology Reports* **35**, 81–88 (2016).
10. Ma, L., Ballantyne, C., Brautbar, A. & Keinan, A. Analysis of Multiple Association Studies Provides Evidence of an Expression QTL Hub in Gene-Gene Interaction Network Affecting HDL Cholesterol Levels. *PLOS ONE* **9**, e92469 (2014).

REVIEWERS' COMMENTS:

Reviewer #1 (Remarks to the Author):

Very helpful responses to questions raised on review. Could more of the specific responses and data be included in the MS, possibly as supplementary information?

No major concerns and all points addressed.

James Brenton

Reviewer #2 (Remarks to the Author):

The authors have addressed previous comments by including additional data & analysis as well as more detailed explanations.

REVIEWERS' COMMENTS:

Reviewer #1 (Remarks to the Author):

Very helpful responses to questions raised on review. Could more of the specific responses and data be included in the MS, possibly as supplementary information?

Many thanks for your review, we have now added some of the data provided on our responses as supplementary data in the manuscript. No major concerns and all points addressed. James Brenton

Reviewer #2 (Remarks to the Author): The authors have addressed previous comments by including additional data & analysis as well as more detailed explanations.

Many thanks for your review.